# Post Placement and Restoration of Endodontically Treated Canines: A Finite Element Analysis Study

**DOI:** 10.3390/ijerph19158928

**Published:** 2022-07-22

**Authors:** Alexandru Dan Popescu, Dragoș Laurențiu Popa, Andreea Gabriela Nicola, Ionela Teodora Dascălu, Cristian Petcu, Tiberiu Tircă, Mihaela Jana Tuculina, Horia Mocanu, Adela Nicoleta Staicu, Lelia Mihaela Gheorghiță

**Affiliations:** 1Department of Endodontics, Faculty of Dental Medicine, University of Medicine and Pharmacy of Craiova, 200349 Craiova, Romania; alexandrudanpopescu20@gmail.com (A.D.P.); cristipetcu80@yahoo.com (C.P.); cenusoiu.adela@yahoo.com (A.N.S.); leliagheorghita@yahoo.com (L.M.G.); 2Department of Automotive, Transportation and Industrial Engineering, Faculty of Mechanics, University of Craiova, 200478 Craiova, Romania; 3Department of Oro-Dental Prevention, Faculty of Dental Medicine, University of Medicine and Pharmacy of Craiova, 200349 Craiova, Romania; andeea_anghel@yahoo.com (A.G.N.); tiberiu.tirca@yahoo.com (T.T.); 4Department of Orthodontics, Faculty of Dental Medicine, University of Medicine and Pharmacy of Craiova, 200349 Craiova, Romania; marceldascalu@yahoo.com; 5Faculty of Dental Medicine, University Titu Maiorescu of Bucharest, 031593 Bucharest, Romania; horia.mocanu@prof.utm.ro

**Keywords:** devital tooth, fiberglass post, metal post, caries restoration

## Abstract

The purpose of this study was to show the improved outcomes of restoring endodontically treated teeth with fiberglass posts compared to restorations using metal posts. In our study, we used the Finite Element Method (FEM), which is based on the principle that a physical model that supports a given load distributes the stress throughout its volume. We sought to assess what stress results in a tooth when it is restored using a fiberglass post compared to restoration using a metal post. The finite element analysis showed that a system consisting of a tooth with a fiberglass post is more stable in terms of the maximum stress than a system consisting of a tooth with a metal post. The maximum displacements and deformations were obtained in the case of a canine restored with a fiberglass post, which showed that this system had a high elasticity, therefore, higher strength than a canine restored with a metal post, which had high rigidity.

## 1. Introduction

During the exercise of the functions of the dento-maxillary apparatus, structural changes occur in the teeth, which may be physiological or pathological. Of the physiological changes, attrition is one significant process, while large losses of dental hard substance can occur in carious lesions or as a result of dental fractures. Failure to properly treat these lesions can lead to loss of tooth vitality, causing serious dental restoration problems [1].

As a result of a large loss of dental hard substance, the tooth structure loses its elasticity, and the tooth has lower mechanical strength. A devital tooth may crack or even fracture under high masticatory force [2,3]. On the other hand, bulky and incorrectly adapted dental restorations may result in marginal microleakage, leading to reinfection of the endodontic space or fracture of the tooth [4].

In order to perform the correct endodontic treatment, an appropriate modern chemomechanical treatment that helps to achieve mechanical, chemical, and microbiological goals must be applied, creating a clean endodontic space. Radiographs and CBCTs allow an accurate and complete diagnosis of periapical pathology and also provide important information on the success of endodontic treatment [5,6,7,8].

A frequently used method in the restoration of devital teeth is the filling of teeth with root posts [9,10]. There is a wide range of posts, with both metal and non-metal posts being used in practice [11].

Numerous studies have shown that there are no major differences between fiberglass and metal posts, unless the restored tooth had two or fewer remaining crown walls. Fiberglass posts have been shown to last better in the medium to long term (3 to 7 years) [12].

In the use of both fiberglass and metal posts, there is a considerable loss of dental hard substance at the root level where the post is placed. Both fiberglass and metal posts have a higher modulus of elasticity than dentin, which increases the risk of fractures [13,14,15,16].

In addition to the loss of dental hard substance resulting from root canal preparation, the different mechanical properties of the posts compared to the tooth properties can cause a decrease in tooth strength [17,18,19,20,21].

There are several characteristics by which we can assess the quality of a post. These are strength and retention in the endodontic space, preservation of tooth structure, the Ferrule effect, and retrievability. The retention of the post in the canal depends on the length of the post, the type of cement used, and the diameter of the post. The way the cement is applied plays an important role in post retention when a regular adhesive resin is used in the root canal. Studies have shown that cements based on self-adhesive resins are less sensitive to cementing processes than cements based on regular resins [22,23].

Other methods for both coronal and root reconstruction of devital teeth have also been researched, but no ideal alternative has been found, and there is always compromise [24,25,26,27]. As a result of the loss of dental vitality, changes in the color of the devital teeth can occur, i.e., they are much darker in color than vital teeth. Devital teeth can become unsightly, requiring prosthetic restoration [28].

## 2. Materials and Methods

In our study, we used the Finite Element Method (FEM), which is based on the principle that a physical model that supports a given load distributes stress throughout its volume. The distribution of these stresses and their magnitude and orientation depend not only on the location of the loads, the volume, and the three-dimensional geometry of the model under study but also on the material properties. At the same time, the stresses are influenced by the interaction between the analyzed model and the external environment, and they can be calculated by mathematical models. In these mathematical models, the aspects of the physical model under analysis, such as the arrangement and size of loads, geometry, material properties, boundary, and contact conditions are modeled by mathematical functions. These mathematical systems can be precise or approximate, depending on the level of precision required and based on experimental determinations and analysis. For the determination of stress, in the solution method, the description structure is included in equations that are based on the theory of continuum solid mechanics, these being solved only by mathematical symbolism and methods, often using different approximations [29].

### Analysis by Finite Element Method of the Dento-Maxillary System and an Intact Canine 3.3

In order to obtain a simulation as close to reality as possible, a model of canine 3.3 containing enamel, dentin, and pulp was virtually implanted in the jawbone of the analyzed system (Figure 1).

This model was exported into the finite element analysis program Ansys Workbench, and its interface is shown in Figure 2. The SolidWorks program has a command that works as a link to Ansys, so the defined geometry can be exported automatically. It was intended that the maxilla be replaced by the reaction force that produces the contact force in the lifting motion. Thus, in the Ansys program, the maxilla was completely suppressed.

It is also very important that the materials attached to the virtual models of the components of the dento-maxillary apparatus are correctly established and defined in the analysis program. Thus, the selective bibliography was reviewed [30,31,32,33], generating a library with all the materials needed for simulation with finite elements: cortical bone, dentin, enamel, and pulp.

The surfaces on which the forces acted were those determined as contact surfaces in the kinematic and dynamic simulation of the lifting movement of the mandible. The actual one-second evolution of the force obtained in an Excel Office file was also used. This temporal evolution was copied directly into the table of force in Ansys Workbench, as shown in Figure 3.

The analyzed system was divided into finite elements using the automatic algorithm of Ansys. With all these elements defined, it was possible to run the application and obtain maps of the results, including the stress map, displacement map, and deformation map.

## 3. Results

### 3.1. Analysis by Finite Element Method of the Dento-Maxillary System and Canine 3.3 with a Nickel–Chromium Alloy Post

In order to generate the model of the canine with a post, the models of the components of this microsystem were generated one by one using CAD methods and techniques. These components were loaded into the Assembly module where, using CAD techniques, the microsystem of canine 3.3 with a pivot was obtained (Figure 4). This system was artificially implanted into the maxilla of the dento-maxillary system, as was also performed with the intact canine.

The entire dento-maxillary apparatus was automatically exported to the Ansys Workbench. Figure 5 shows the mandible and also the microsystem of canine 3.3 with a post, after its artificial isolation.

New materials were defined and loaded into the Engineering Data module: gutta-percha, ceramic micro + protection, and Ni + Cr post.

After running the application, result maps were obtained. Figure 6 shows the stress maps of the 3.3 canine and its components. The maximum stress value was 3.0937 × 10^5^ Pa.

Figure 7 shows the displacement maps of the canine and its components. The maximum displacement was 3.6524 × 10^−7^ m.

Figure 8 shows the deformation maps of the analyzed elements. The maximum deformation value was 3.5735 × 10^−5^ m/m, which was recorded at the distance from the area of the dental cervical, which is considered the normal flexion zone of the tooth. Cast metal root canal restorations have a high modulus of elasticity compared to root dentin, creating a stiffer restorative complex that causes high stress concentrations on the root.

### 3.2. Analysis by Finite Element Method of the Dento-Maxillary System and Canine 3.3 with a Glass Fiber Post

For this analysis, the previous model was used, updated with the materials: gutta-percha, ceramic micro protection, fiberglass post, and cement.

After running the application, the result maps were obtained. Figure 9 shows the stress maps of the canine 3.3 components. The maximum stress value was 3.66 × 10^5^ Pa, which was higher than that of the metal post restoration.

Figure 10 shows the displacement maps of the canine 3.3 components. The maximum displacement was 4.8315 × 10^−7^ m, which was much higher than that of the metal post restoration.

Figure 11 shows the deformation maps of the canine 3.3 components. The maximum deformation was 4.2159 × 10^−5^ m/m. This value was higher than that of the metal post restoration; however, the pressure was exerted apically and in the dental cervical zone; which are the functional areas of the tooth. The similar modulus of elasticity between the fiberglass post and dentin led to a reduced stress concentration and restored the stress distribution to a state similar to that of a healthy tooth.

## 4. Discussion

In recent years, numerous studies have been carried out on both the advantages and disadvantages of using non-metallic pivots, including fiberglass, carbon fiber, and fiberglass reinforced composite resin (FRC) posts. Fiberglass posts can be used in the conventional way, following the model of metallic posts [34,35] or using the concept of a single post on which the resinous material abutment will be built. Unfortunately, failures also occur when using fiberglass posts due to the material from which the post is made, the way the post is placed in the root canal, or due to faulty post cementation techniques. Secondary caries can most often be seen at the site of failure [35,36]. This treatment error can be avoided if the Ferrule effect is observed.

Until the late 1980s, metal alloys, such as nickel–chromium alloys, were widely used by dentists. They had some benefits; during retreatment they were easy to replace, and after casting, the post benefits from a metal abutment to reconstruct the crown part. The difficulty of placement in the root canal, the frequent occurrence of vertical root fractures, and the need for some laboratory steps for fabrication have meant the use of metal alloys has decreased, and with time they have been replaced by non-metal posts.

Fiberglass posts have significant benefits in terms of appearance, especially in the front teeth. Carbon fiber posts, although offering the same qualities as fiberglass posts, have a much darker color [37]. The modulus of elasticity of fiberglass posts, demonstrated in our study by generating component models using CAD methods and techniques in the Engineering Data module, was lower than that of Ni-Cr alloy posts; therefore, they have a much higher elasticity than metallic posts, which are much stiffer.

The stresses propagate to the entire tooth structure (enamel, dentin, pulp chamber, and cement), periodontium, and bone structures [38,39]. Vital teeth are under continuous stress, and this is largely due to the function of the masticatory system and the quality of occlusion [11,39]. Through finite element analysis, a comparison was made between the stress resistance of an intact vital tooth and an endodontically treated tooth. The forces applied to both an intact molar and an endodontically treated molar caused damage to the occlusal surface, generating cusp fractures, with the difference being the time of occurrence [39]. The results obtained in our study showed the maximum stress exerted on the endodontically treated canine 3.3 was at the level of the dental cervical area, generating fractures. Therefore, we can say that an endodontically treated tooth, regardless of the materials or methods used, will have a lower strength compared to an intact tooth due to the loss of dental hard substance. Endodontic research over the past decade has contributed to the development of instruments that provide optimal treatment results, even for teeth with atypical morphology. However, there are situations in which the dental hard substance cannot withstand the tension of the instruments and fails, propagating along the entire length of the root canal and leading to the appearance of microcracks in the apical, middle, and coronal thirds, as could be seen in the finite element analysis [40].

The geometric pattern of the endodontic instruments, the technique used, and the shape of the root canal (straight or curved) are important factors in achieving optimal treatment. As mentioned above, the root canal has three portions (coronal, middle, and apical) and, according to studies conducted on a tooth with a straight root canal, the highest stress occurs in the apical portion [41,42], constituting a disadvantage in the endodontic treatment of canine 3.3. There must be a close connection between the materials used for the treatment to give an optimal result. Thus, the use of a thick fiberglass post at the entrance of the root canal, obtaining a small space between the dentinal walls and the post when inserting it into the canal and perfect adhesion at the level of the abutment, offers the patient a quality and long-lasting dental restoration [36,43,44].

Several researchers have obtained conflicting results regarding dental restorations, with some preferring a full resin restoration, and others opting for converting the actual tooth to an abutment and full coverage of it with a physiognomic crown [45,46,47]. Moreover, the decision is also up to the patients, especially with regard to their preferences when it comes to front teeth, as achieving a quality physiognomic effect is important.

Other theories claim that restoring the tooth using composite resins gives the tooth approximately the same properties it had when it was free of damage, without the need for the use of fiberglass-reinforced composite resins. Premolars restored with fiberglass-reinforced composite resins are susceptible to fractures in the dental cervical areas compared to those restored occlusally with either simple or reinforced composite resins [48,49]. However, a certain class of fiber-reinforced composite resin (SFRC-everX Flow) has been subjected to in vitro experiments and shows greater resistance to fractures. Unfortunately, it also has the disadvantage that it shrinks more during polymerization [50,51,52].

In contrast to these studies, Vishanth Subashri and co-workers conducted an in vitro study on maxillary incisors with significant loss of dental hard substance (approximately 50%), restoring these teeth using different techniques, and concluding that composite coronal–radicular restorations have the highest resistance to fracture compared to direct composite restorations. Following this study, they stated that the tandem use of fiberglass posts and direct composite restorations resulted in the worst results for resistance to fracture [53].

An ideal post that approximates the structure of physiological dental dentin would be easy to insert, would not require large sacrifices of dental hard substance in the preparation of the radicular canal, would make the stresses during masticatory functions evenly distributed, and would be as aesthetically pleasing as possible [54,55]. So far, the closest is the fiberglass post, which fulfils many of the characteristics of the ideal post and has a modulus of elasticity similar to that of the tooth [56].

Excellent results have been obtained from in vitro experiments using dentin posts, which are a very good alternative in the restoration of endodontically treated teeth due to the fact that the stresses occurring in the post are distributed over its entire length, unlike fiber posts, where the stresses are distributed unevenly [57].

An important step in the successful completion of endodontic treatment using fiber posts is cementation. This process is intensely debated in the specialized literature, with many authors trying different protocols to increase the adhesion rate of the post in the dentin of the tooth in question. Samah Saker and Mutlu Özkan concluded that the cementation of fiber-reinforced posts in the root canal is dependent not only on the cement used but also on the treatment of the root canal dentin, with the retention of fiber-reinforced resin posts being higher in teeth treated with EDTA solution (17%) than in teeth treated with phosphoric acid (37%) [58].

Following the restoration of endodontically treated teeth, both with fiber posts and metal posts (Ni-Cr in our case), a very important aspect for the survival rate of the tooth on the arch is to cover it with a physiognomic crown. The crown must protect the remaining structure of the tooth and at the same time maintain the health of the periodontium [59].

Statistically, the survival and failure rates of endodontically treated teeth with fiberglass or metal posts are not significantly different. In 2022, Nino Tsintsadze et al. considered a total of 188 studies and concluded that both materials can be used with confidence in performing treatment when a significant amount of coronary structure is missing, as the success rate is 92.8% for fiberglass posts and 78.1% for metallic posts [60,61,62].

## 5. Conclusions

Finite element analysis showed that a system consisting of a tooth with fiberglass was more stable in terms of maximum stresses in comparison to a system consisting of a tooth with a metal post; therefore, the fiberglass was more resistant to stress.

The fact that the maximum displacements and deformations were obtained in the case of the canine restored with a fiberglass post showed that this system had high elasticity, compared to when the canine was restored with a metal post, which had high rigidity.

Fiberglass posts showed high stresses in the cervical region due to their flexibility but also in the apical area, which would lead to a reduced risk of the root fracture of endodontically restored and treated teeth.

## Figures and Tables

**Figure 1 ijerph-19-08928-f001:**
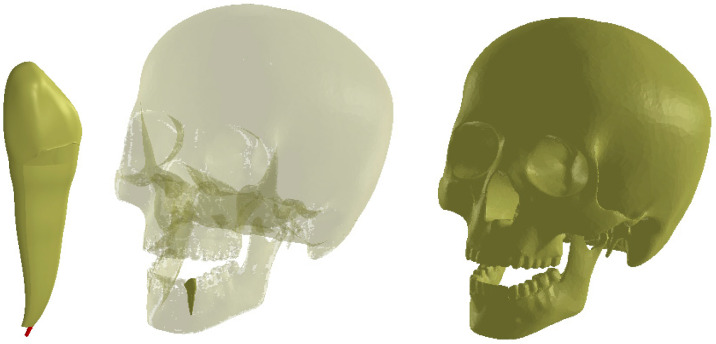
Model of the dento-maxillary system with intact canine 3.3.

**Figure 2 ijerph-19-08928-f002:**
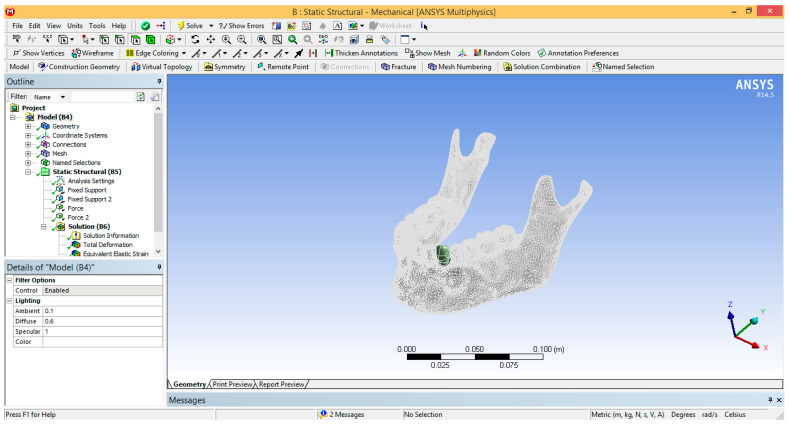
Ansys Workbench interface.

**Figure 3 ijerph-19-08928-f003:**
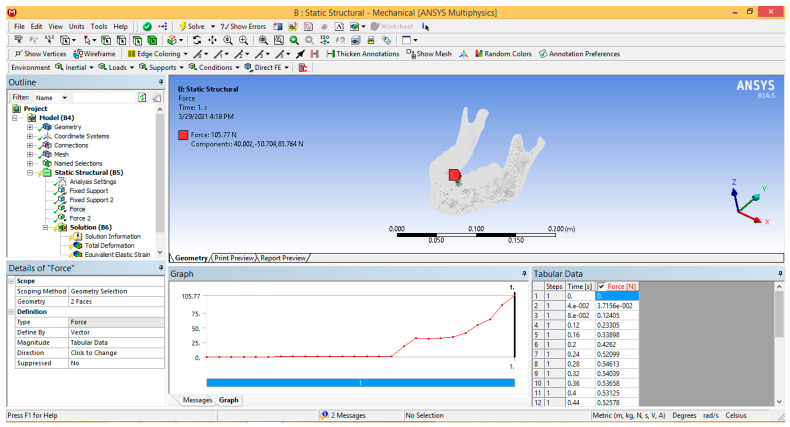
Defining forces in Ansys Workbench.

**Figure 4 ijerph-19-08928-f004:**
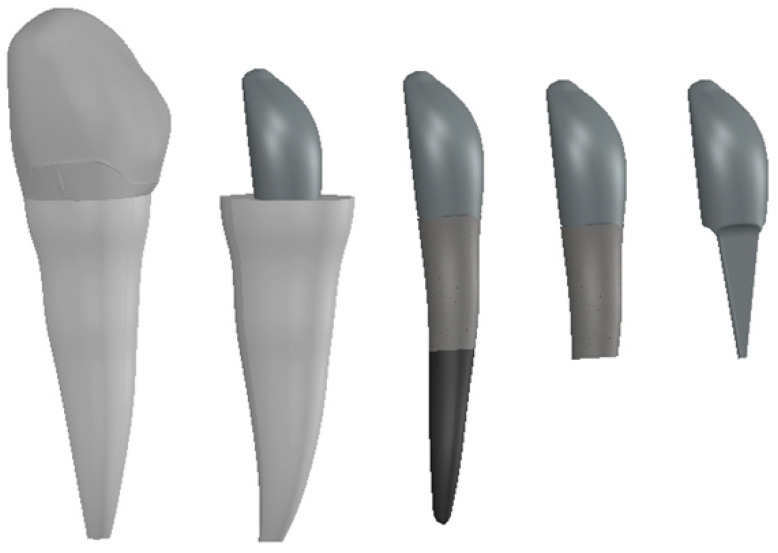
The model of the microsystem of canine 3.3 with a post.

**Figure 5 ijerph-19-08928-f005:**
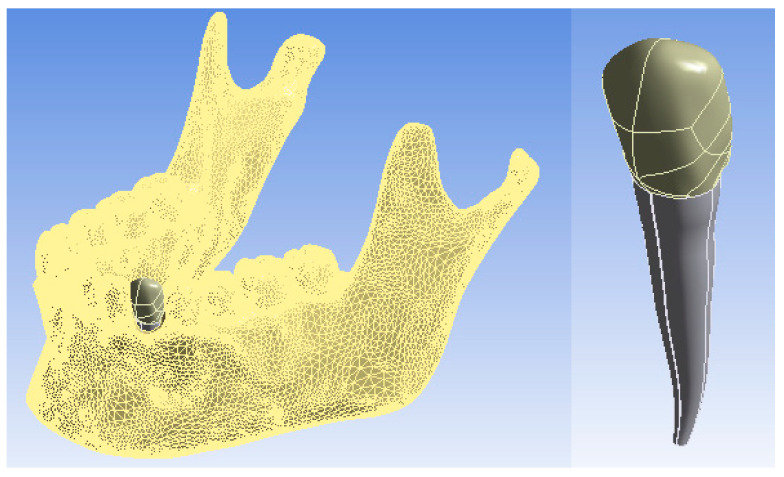
The whole system and the microsystem of canine 3.3 with a post in Ansys.

**Figure 6 ijerph-19-08928-f006:**
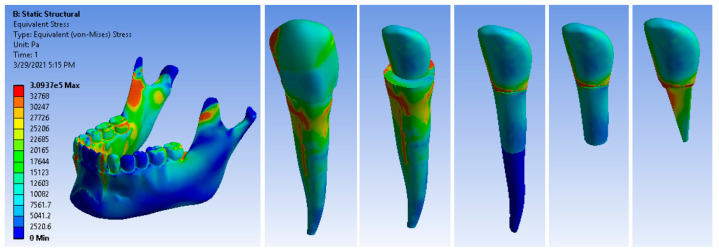
Maps of stress generated in the root with a Ni + Cr post.

**Figure 7 ijerph-19-08928-f007:**
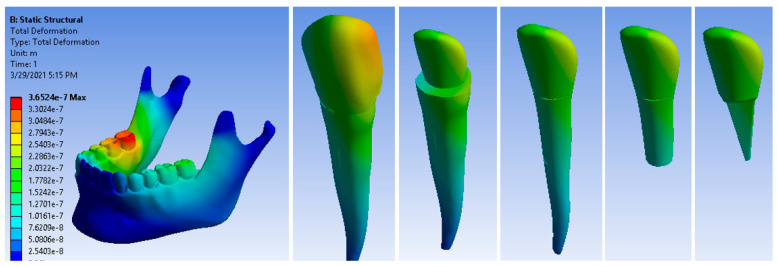
Displacement maps of a restored canine with a Ni + Cr post, under masticatory forces.

**Figure 8 ijerph-19-08928-f008:**
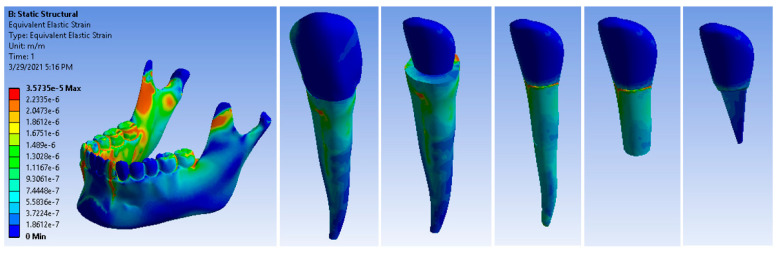
Maps of deformation under stress of a canine restored with a Ni + Cr post.

**Figure 9 ijerph-19-08928-f009:**
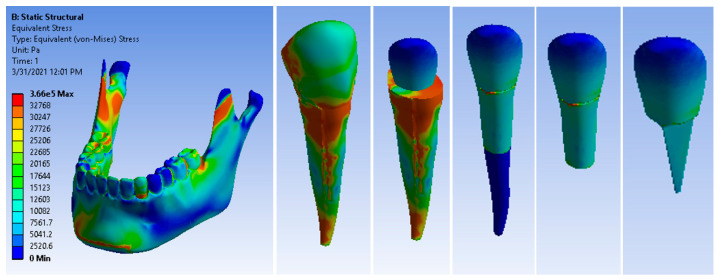
Maps of stress generated in the root by a glass fiber post.

**Figure 10 ijerph-19-08928-f010:**
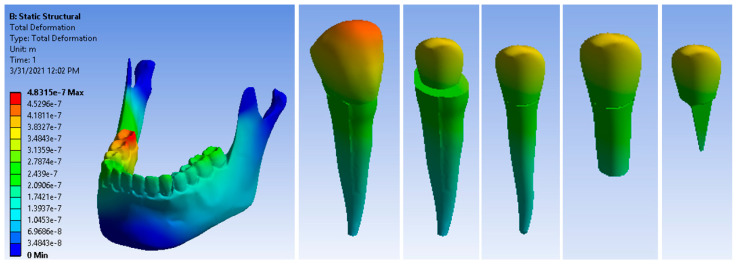
Displacement maps of a restored tooth with a fiberglass post under masticatory forces.

**Figure 11 ijerph-19-08928-f011:**
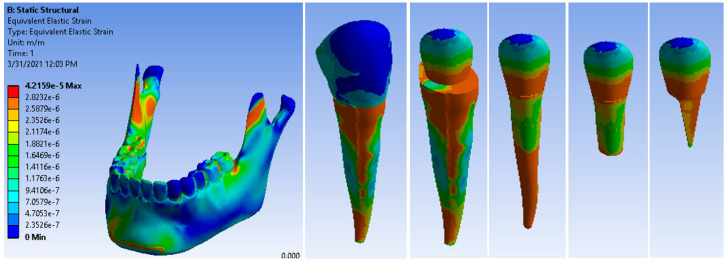
Maps of deformation under stress of a canine restored with a fiberglass post.

## Data Availability

The authors declare that the data from this research are available from the corresponding authors upon reasonable request.

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
