# Peer review of "Post Placement and Restoration of Endodontically Treated Canines: A Finite Element Analysis Study"

_ijerph, 2022, doi:10.3390/ijerph19158928_

Round 1

Reviewer 1 Report

In this study, the authors used finite elements method(FEM) to investigate the maximum stress of a canine restored with a fiberglass post compared to a canine restored with a metal post. The results of this study show that fiberglass post yields a high elasticity while the metal post obtains a high rigidity. Although the novelty of this study is not high and  the conclusion is not beyond the common sense, the authors do an The comments of this reviewer is as follows: 

 — The title of this manuscript should be toned town. The authors only test a canie instead of other critical teeth such as incisor, premolar and molar.

 — what is the limitation of this study ? Are the results of a FEM study clinically relevant ?

 — The introduction and discussion part should be improved. Please cite classic and strong studies. 

— Some figures such as Figure 11 should be adjusted. 

Author Response

Thank you for your review and time!

We have changed the title of the article to “Post Placement and Restoration of Endodontically Treated Canines: A Finite-Element Analysis Study”, replacing the word “Teeth” with “Canines” as you suggested;

-          The study has limitations because it was performed only on the canines but it has clinical significance for the reasons that it guides doctors in organizing and choosing the treatment plan in case of restoration of devital teeth;

-          We’ve improved the introduction by removing the sequence “The success rate is about 97% for a correctly performed endodontic treatment, but the long-term prognosis of a devital tooth is also influenced by the correctness of the tooth restoration, i.e. the type of restoration and the materials used.” and adding references suggested by MDPI Board : “Bergenholtz, G. Assessment of treatment failure in endodontic therapy. J. Oral Rehabil. 2016, 43, 753-8.”(reference no. 5);

-          We’ve also improved the discussions by citing strong new sources of recent studies conducted in 2022 – reference no. 60 (Tsintsadze, N.; Margvelashvili-Malament, M.; Natto, Z.S.; Ferrari, M. Comparing survival rates of endodontically treated teeth restored either with glass-fiber-reinforced or metal posts: A systematic review and meta-analyses. J. Prosthet. Dent. 2022, 13,00047-6.), 2020 – reference no. 61 (Sarkis-Onofre, R.; Amaral Pinheiro, H.; Poletto-Neto, V.; Bergoli, C.D.; Cenci, M.S.; Pereira-Cenci, T. Randomized controlled trial comparing glass fiber posts and cast metal posts. J. Dent. 2020, 96, 103334.) and 2017 – reference no. 62 (Ahmed, S.N.; Donovan, T.E.; Ghuman, T.; Survey of dentists to determine contemporary use of endodontic posts. J. Prosthet.Dent. 2017, 117, 642-645.);

-          And lastly, we modified the figures and the layout on the page according to the journal template.

Reviewer 2 Report

Dear Authors, 

you made a great work! however, some improvements are mandatory before acceptance. 

Author Response

Thank you for your review and time!

We improved the English language by using the translation services offered by MDPI, thus improving the transaltion of the article.

 Note: Within the article, a statistical analysis has not been done yet but we consider to do it in a later study.

Round 2

Reviewer 1 Report

The quaility of this manuscript has been improved after revison.